# Endoscopic Ultrasound-Guided Gallbladder Drainage: Beyond Cholecystitis

**DOI:** 10.3390/diagnostics13111933

**Published:** 2023-06-01

**Authors:** Nicholas J. Koutlas, Swati Pawa, Greg Russell, Rishi Pawa

**Affiliations:** 1Department of Medicine, Section on Gastroenterology, Wake Forest School of Medicine, Winston-Salem, NC 27157, USA; nkoutlas@wakehealth.edu (N.J.K.); spawa@wakehealth.edu (S.P.); 2Department of Biostatistics and Data Science, Wake Forest School of Medicine, Winston-Salem, NC 27157, USA; grussell@wakehealth.edu

**Keywords:** endoscopic ultrasound, lumen apposing metal stents, cholecystitis, cholelithiasis, malignant biliary obstruction

## Abstract

Endoscopic ultrasound-guided gallbladder drainage (EUS-GBD) is an alternative to surgery for acute cholecystitis (AC) in poor operative candidates. However, the role of EUS-GBD in non-cholecystitis (NC) indications has not been well studied. We compared the clinical outcomes of EUS-GBD for AC and NC indications. Consecutive patients undergoing EUS-GBD for all indications at a single center were retrospectively analyzed. Fifty-one patients underwent EUS-GBD during the study period. Thirty-nine (76%) patients had AC while 12 (24%) had NC indications. NC indications included malignant biliary obstruction (*n* = 8), symptomatic cholelithiasis (*n* = 1), gallstone pancreatitis (*n* = 1), choledocholithiasis (*n* = 1), and Mirizzi’s syndrome (*n* = 1). Technical success was noted in 92% (36/39) for AC and 92% (11/12) for NC (*p* > 0.99). The clinical success rate was 94% and 100%, respectively (*p* > 0.99). There were four adverse events in the AC group and 3 in the NC group (*p* = 0.33). Procedure duration (median 43 vs. 45 min, *p* = 0.37), post-procedure length of stay (median 3 vs. 3 days, *p* = 0.97), and total gallbladder-related procedures (median 2 vs. 2, *p* = 0.59) were similar. EUS-GBD for NC indications is similarly safe and effective as EUS-GBD in AC.

## 1. Introduction

More than 20 million people, including 7.9% of males and 16.6% of females, are impacted by gallstone-related diseases in the United States [1]. The majority remain asymptomatic and require no treatment. Meanwhile, upwards of 20% of patients with cholelithiasis develop symptoms that comprise biliary colic, cholecystitis, choledocholithiasis, cholangitis, and pancreatitis. For those with symptoms, laparoscopic cholecystectomy remains the standard of care [2]. Each year, roughly 600,000 patients undergo laparoscopic cholecystectomy, making it the sixth most common operation performed [3,4]. However, surgical management may be precluded in select patients due to comorbidities, intra-abdominal inflammation, or hemodynamic instability.

Traditionally, the primary alternative to surgery for acute cholecystitis (AC) has been percutaneous transhepatic gallbladder drainage (PT-GBD) [5,6,7,8]. PT-GBD is highly effective, widely available, and can be performed with only local anesthesia if needed [9,10,11]. However, PT-GBD has several notable adverse effects which include bleeding, bile leak, insertion site discomfort, frequent need for re-interventions, and a decrease in patient quality of life. Furthermore, cholecystitis can still recur in up to 47% of patients. For these reasons, patients almost invariably prefer internal drainage when feasible [12].

Endoscopic ultrasound-guided gallbladder drainage (EUS-GBD) using plastic double pigtail stents was first reported in 2007 by Baron and Topazian in a poor surgical candidate with acute cholecystitis [13]. EUS-GBD with a lumen-apposing metal stent (LAMS) was first described in 2012 by Itoi et al. [14]. The application of LAMS for EUS-GBD has drastically simplified the procedure as stent insertion can be performed in just one step. Numerous studies have now demonstrated EUS-GBD with LAMS to be safe and effective in poor operative candidates [15,16,17,18]. In a recent systematic review and meta-analysis, EUS-GBD with LAMS demonstrated rates of technical success, clinical success, and adverse events of 94.7%, 92.1%, and 11.7%, respectively [19]. When compared to PT-GBD, EUS-GBD has been shown to have similar technical and clinical success with fewer adverse events, re-interventions, readmissions, and recurrences of cholecystitis [20].

Additional applications for EUS-GBD beyond AC have increasingly been explored. These include symptomatic cholelithiasis, secondary prevention of gallstone-related disease, and rescue treatment of malignant biliary obstruction (MBO) [21,22,23]. While current data support EUS-GBD as the preferred alternative to surgery for AC in high-risk surgical patients, less is known about its utility for these newer indications. As such, we aim to compare clinical outcomes of EUS-GBD for AC and non-cholecystitis (NC) indications.

## 2. Materials and Methods

### 2.1. Study Design, Subjects, and Data Collection

Consecutive non-surgical patients undergoing EUS-GBD with a LAMS for all indications from July 2019 to October 2022 at a single tertiary care center were retrospectively analyzed. Non-surgical patients were deemed by surgery to be poor surgical candidates due to medical comorbidities or extensive intra-abdominal inflammation. A manual review of the electronic health record was performed by viewing procedure reports, inpatient and outpatient provider notes, laboratory studies, and imaging reports. Data on patient demographics, anticoagulant/antiplatelet use, procedure-related details, and pertinent clinical outcomes were collected. Data recorded were stored in a secured database in accordance with our local institutional review board (IRB number: 00077330). Patients were grouped into one of two cohorts based on the indication for EUS-GBD: acute cholecystitis (AC) and non-cholecystitis (NC). NC indications included symptomatic cholelithiasis, secondary prevention of gallstone-related disease, and patients with MBO (with a patent cystic duct) in whom transpapillary and EUS-guided biliary drainage were unsuccessful.

### 2.2. Outcomes

The primary outcome of our study was technical success defined by the appropriate positioning of the LAMS in the gallbladder lumen confirmed by endoscopic, sonographic, and fluoroscopic images. Secondary outcomes included a total number of gallbladder interventions (including index procedure), procedure duration, post-procedure hospital length of stay, and adverse events. Adverse events were defined as early (within 4 weeks) or delayed (beyond 4 weeks) and classified according to the lexicon created by the American Society of Gastrointestinal Endoscopy [24].

Clinical success for patients with symptomatic cholelithiasis and secondary prevention of gallstone-related disease was defined by the absence of biliary colotic and future gallstone-induced complications requiring hospital readmissions. For patients with MBO, a decrease in serum bilirubin by at least 50% at two weeks post-procedure indicated clinical success.

### 2.3. Statistical Analysis

Categorical variables were reported as frequencies and compared using Fischer’s exact test. Continuous variables were reported as medians with interquartile ranges (IQR) and Wilcoxon two-sample test was used for significance testing. Survival after EUS-GBD was compared between AC and NC patients using a log-rank test with censoring performed on the date of the last follow-up. A *p*-value of <0.05 was used to determine statistical significance.

### 2.4. Procedure Detail

All procedures were performed under general anesthesia. A linear echoendoscope (GF-UTC 180; Olympus, Tokyo, Japan) was used to visualize the gallbladder and identify a site in the gastric antrum or duodenal bulb with an avascular path (Figure 1). A 22-gauge needle (EchoTip Ultra, Cook Medical, Winston-Salem, NC, USA) was used to puncture the gallbladder (Figure 2a). Bile was then aspirated to confirm the needle was appropriately placed in the gallbladder. Following this, a dilute contrast injection was performed to distend the gallbladder lumen (Figure 2b). In patients with MBO, the presence of contrast in the common bile duct confirmed cystic duct patency.

An electrocautery-enhanced LAMS (Axios, Boston Scientific, Marlborough, MA, USA) was then introduced into the gallbladder. Distal flange deployment was performed under EUS guidance, and the proximal flange was deployed with direct endoscopic visualization (Figure 2c,d). Following LAMS deployment, a 0.025 inch in diameter and 450 cm in length straight-tip Visiglide 2 wire (Olympus, Tokyo, Japan) was inserted into the gallbladder. The LAMS was dilated under endoscopic and fluoroscopic guidance using a CRE wire-guided balloon (Boston Scientific, Marlborough, MA, USA) up to the diameter of the LAMS (Figure 3a). A 7 Fr by 4 cm plastic double pigtail (DPT) stent was then placed through the LAMS into the gallbladder (Figure 3b).

Post-procedure, all patients remained nil per os overnight and received intravenous hydration. Antibiotics were continued in patients who underwent EUS-GBD for AC. Patients were initiated on a clear liquid diet the following day and advanced as tolerated. Esophagogastroduodenoscopy (EGD) with cholecystoscopy was performed four to six weeks after initial LAMS placement. In patients with gallstone-related disease, we routinely removed stones prior to LAMS removal. This was achieved with saline lavage, stone extraction baskets, or electrohydraulic lithotripsy (EHL). Following LAMS and DPT stent removal, a 7 Fr × 4 cm DPT stent was left in the gallbladder lumen for long-term drainage.

## 3. Results

### 3.1. Patient Characteristics

A total of 51 patients underwent EUS-GBD during the study period. The median age was 73 years (Interquartile range 62.5, 81.5) and 37% of patients were female. The median Charlson Comorbidity Index (CCI) was 7 (IQR 5, 8.5). Thirteen (25%) patients were taking anticoagulants prior to the procedure. Eight (16%) patients, all in the AC group, had indwelling percutaneous cholecystostomy tubes at the time of the index procedure.

Of the 51 patients, 39 (76%) underwent EUS-GBD for acute cholecystitis. Twelve (24%) patients had non-cholecystitis indications. These included eight patients with MBO and one patient each with symptomatic cholelithiasis, cholelithiasis and choledocholithiasis, gallstone pancreatitis, and Mirizzi’s syndrome. There was no difference in age, gender, CCI, and anticoagulant use between the two cohorts (Table 1).

### 3.2. Procedure Variables

Procedure details are illustrated in Table 2. The LAMS was placed via a transduodenal route in 62% of patients with the majority (83%) undergoing drainage with a 10 mm LAMS. The route of drainage and size of LAMS was not significantly different between the two groups. A plastic DPT stent was placed within the LAMS during the index procedure in 86% of AC patients and 91% of NC patients (*p* > 0.99).

LAMS were removed in 78% (28/36) of AC patients compared to 55% (6/11) in NC patients (*p* = 0.25). The LAMS was left in place in eight patients in the AC group and five in the NC group due to patient preference in the setting of advanced medical disease with an estimated survival of fewer than three months. The median LAMS duration prior to removal was 35 days (IQR 28, 49) and 30 days (IQR 27, 56) in the AC and NC cohorts, respectively (*p* = 0.90). At the time of LAMS removal, a DPT stent was replaced in 93% of AC patients and 100% of NC patients (*p* > 0.99). A DPT stent was not replaced in two patients. In the first patient, the gallbladder was shriveled and contracted, making stent placement technically difficult. The second patient developed a cholecystocolonic fistula and hence DPT placement was not pursued to avoid mechanical trauma, thereby allowing the fistula to heal.

### 3.3. Outcomes

Clinical outcomes of interest are highlighted in Table 3. Overall, technical success was achieved in 47 patients (92%). There was no difference in technical success between the two groups (92% for AC vs. 92% for NC, *p* > 0.99). Technical failures occurred due to inability to locate a safe window for gallbladder puncture (*n* = 2) and stent misdeployment (*n* = 2). In the cases with technical failure, each was successfully treated with endoscopic transpapillary gallbladder drainage. In those with technical success, clinical success was achieved in 94% (34/36) of AC patients and 100% (11/11) of NC patients (*p* > 0.99). Recurrent cholecystitis requiring PT-GBD was the cause of one clinical failure. The second clinical failure had persistent sepsis despite EUS-GBD and ultimately pursued comfort care measures.

The median procedure duration was similar between patients with AC and NC (43 vs. 45 min, *p* = 0.37). Additionally, there was no difference in post-procedure hospital length of stay (median 3 vs. 3 days, *p* = 0.97) between AC and NC. Total gallbladder-related procedures were also similar (median 2 vs. 2 procedures, *p* = 0.59).

Median follow-up duration was 463 days (IQR 92, 572) for AC patients and 131 days (IQR 55, 394) for NC patients (*p* = 0.46). Median survival was 243 days (IQR 39, 347) for AC and 202 days (IQR 48, 287) for NC (*p* = 0.36). None of the deaths were related to EUS-GBD.

### 3.4. Adverse Events

A total of seven (14%) adverse events occurred after EUS-GBD, four in the AC cohort and three in the NC cohort (*p* = 0.33) (Table 4). In the AC group, early adverse events (less than 4 weeks) included post-procedure bleeding (*n* = 2) and abdominal pain resulting in unplanned readmission (*n* = 1). One patient had a delayed adverse event (beyond 4 weeks) and developed a cholecystocolonic fistula (*n* = 1). Of the two cases of post-procedure bleeding, one patient required embolization of the gastroduodenal artery to achieve successful hemostasis. In the second case, bleeding occurred after the initiation of anticoagulation for a newly discovered pulmonary embolus. The patient was managed supportively with one blood transfusion and spontaneous hemostasis was ultimately achieved. The cholecystocolonic fistula was incidentally noted during a planned EGD for transgastric LAMS removal. This was successfully managed with stent removal and closure of the gastric defect through the scope clips. Given the presence of a prior biliary sphincterotomy and a patent biliary tree, the cholecystocolonic fistula closed spontaneously. A colonoscopy was later performed and confirmed the closure of the fistula [25].

All adverse events in the NC cohort occurred within four weeks and included symptomatic pneumoperitoneum (*n* = 2) and post-procedure respiratory failure (*n* = 1). Of the two patients with symptomatic pneumoperitoneum, one occurred after LAMS misdeployment and required exploratory laparotomy on post-procedure day 3 due to a worsening abdominal examination. The second patient with pneumoperitoneum was managed conservatively with intravenous fluids and antibiotics. The case of respiratory failure required re-intubation after the procedure followed by overnight intensive care unit admission.

## 4. Discussion

The safety and efficacy of EUS-GBD in the management of non-surgical patients with acute cholecystitis have been well established. The application of EUS-GBD for non-cholecystitis indications is less well studied. In the present study, we demonstrate that EUS-GBD has similar technical and clinical success for non-cholecystitis indications compared to acute cholecystitis. Additionally, there were no differences in adverse events, procedure duration, post-procedure length of hospital stays, and total gallbladder-related procedures.

Our study noted technical success rates of 92% for both AC and NC patients. These results fit well within the previously reported ranges for technical success for EUS-GBD in AC of 90% to 99% [20,22,26,27,28,29]. Additionally, a prior retrospective study comparing EUS-GBD for AC and NC reported similar findings [23]. In theory, EUS-GBD may be more challenging to perform for NC indications as the gallbladder may not be distended as is typically the case in AC. In addition, the presence of larger gallstones can make distal flange deployment technically challenging. While the results of our study do not support this, it is important to note that the majority of NC patients in our study had MBO rather than non-obstructive gallstone disease.

In this study, clinical success was achieved at similarly high rates for AC and NC at 94% and 100%, respectively. High clinical success rates of EUS-GBD for AC have been frequently reported, with estimates ranging from 89% to 98% [22]. Clinical success for NC indications, however, is not as well described. A case series of 12 patients undergoing EUS-GBD for MBO demonstrated a clinical success rate of 91% [21]. A multicenter retrospective study of 28 patients also reported similar results [30]. Furthermore, a recent systematic review and meta-analysis showed a pooled clinical success rate of 85% for MBO, which is comparable to clinical success rates for EUS-guided biliary drainage [31,32] In a retrospective study of 15 patients undergoing EUS-GBD for all NC indications, 13.3% of patients required admission for biliary disease within one year of EUS-GBD [23] In contrast, we did not have any cases of recurrent biliary disease after EUS-GBD in the NC group. This is likely explained by the standard practice of stone extraction prior to LAMS removal at our institution, a protocol that is not universally performed.

Adverse events occurred in 10% of AC patients and 25% of NC patients in our study, although the difference was not significant. Adverse events have been noted in up to 25% of patients within one year of EUS-GBD for AC [20]. In contrast, a more recent meta-analysis reported a lower rate at 11.7% [19]. For NC indications, Flynn et al. found an adverse event rate of 13.3% within two weeks of EUS-GBD [23]. Furthermore, in patients undergoing EUS-GBD for rescue treatment of MBO, 16% experience adverse events [31]. In our study, the small number of NC patients somewhat limits the interpretation of our results. However, the adverse event rate of 25% in NC patients is within the previously reported range for adverse events of EUS-GBD for AC. As such, it seems likely that EUS-GBD for NC has a similar safety profile as EUS-GBD for AC.

This study has several notable limitations. First, its retrospective design is accompanied by an inherent risk of selection bias. The patients in the NC group may have been preferentially selected for EUS-GBD based on the presence of a distended gallbladder seen on imaging. Patients with less favorable anatomy may have been excluded which could result in overestimation of technical success. Second, the sample size in the NC group is fairly small. As such, our study may not be adequately powered to detect differences in clinical outcomes between the two groups. Lastly, the majority of patients in the NC group had MBO, making it difficult to draw firm conclusions for EUS-GBD in the management of non-cholecystitis gallstone-related diseases.

## 5. Conclusions

In summation, EUS-GBD is a safe and effective method for treating cholecystitis in non-surgical patients. Our data suggest EUS-GBD for non-cholecystitis indications has similar safety and efficacy when compared with patients undergoing drainage for acute cholecystitis. Larger, prospective studies are needed to validate our findings.

## Figures and Tables

**Figure 1 diagnostics-13-01933-f001:**
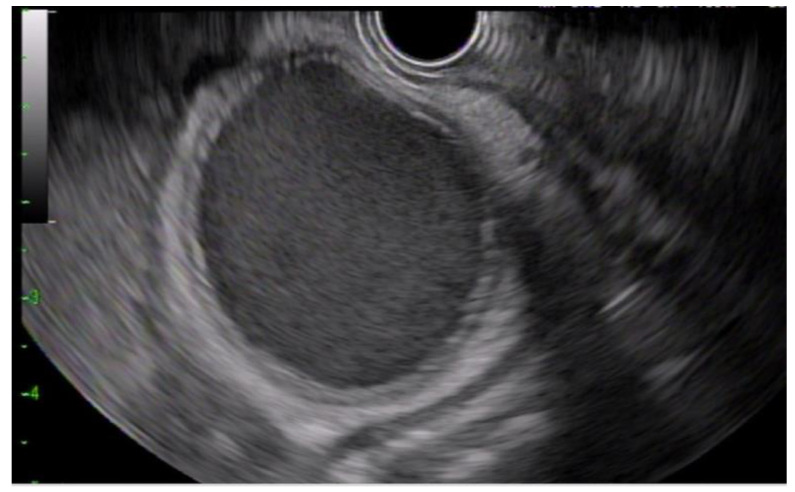
Endoscopic ultrasound image demonstrating a dilated, sludge-filled gallbladder with thickened walls, consistent with acute cholecystitis.

**Figure 2 diagnostics-13-01933-f002:**
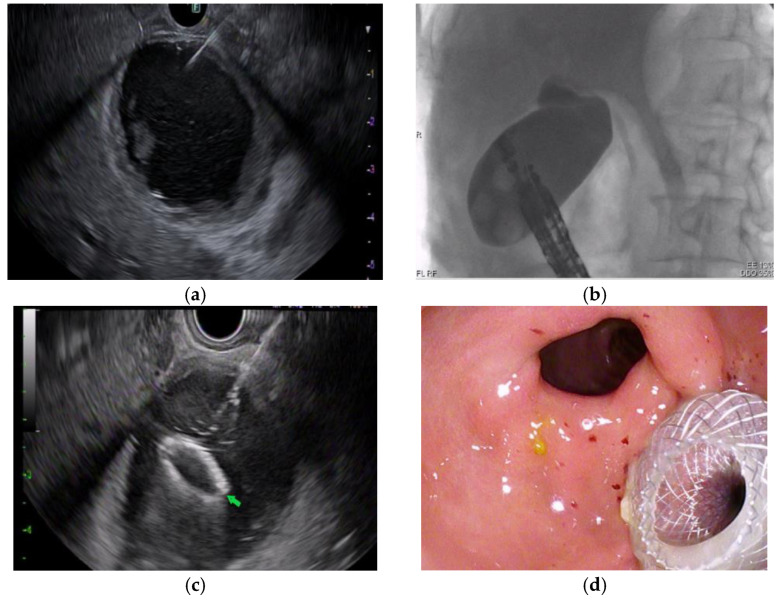
(**a**) Fine needle aspiration of the gallbladder as viewed with endoscopic ultrasound. (**b**) Fluoroscopic image demonstrating contrast filling the gallbladder. (**c**) Distal flange (green arrow) deployment under endoscopic ultrasound guidance. (**d**) Endoscopic image after successful transgastric LAMS placement into the gallbladder.

**Figure 3 diagnostics-13-01933-f003:**
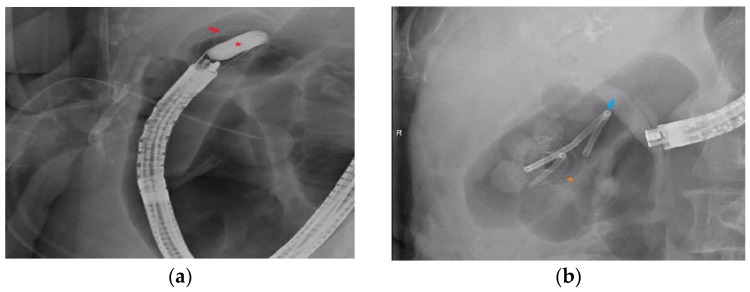
(**a**) Balloon dilation (red star) of lumen apposing metal stent (red arrow). (**b**) Fluoroscopic image of plastic double pigtail stent (blue arrow) after placement through the lumen apposing metal stent (orange star) into the gallbladder.

**Table 1 diagnostics-13-01933-t001:** Characteristics of patients undergoing EUS-GBD.

	Acute Cholecystitis (*n* = 39)	Non-Cholecystitis (*n* = 12)	*p* Value
Age, median (IQR ^1^)	74 (65, 80)	68.5 (61, 89)	>0.99
Gender			
Male	26 (67%)	6 (50%)	0.33
Female	13 (33%)	6 (50%)	
Charlson Comorbidity Index, median (IQR)	7 (5, 8)	7.5 (5, 9.5)	0.75
Anticoagulant use	9 (23%)	4 (33%)	0.47

^1^ Interquartile Range.

**Table 2 diagnostics-13-01933-t002:** Procedure details of EUS-GB for AC and NC indications.

	Acute Cholecystitis (*n* = 36)	Non-Cholecystitis (*n* = 11)	*p* Value
Route of LAMS into gallbladder			
Transgastric	12 (33%)	6 (55%)	0.29
Transduodenal	24 (67%)	5 (45%)	
LAMS size (mm)			
8	3 (8%)	1 (9%)	0.42
10	31 (86%)	8 (73%)	
15	2 (6%)	2 (18%)	
Double pigtail stent within LAMS	31 (86%)	10 (91%)	>0.99
LAMS removed	28 (78%)	6 (55%)	0.25
Double pigtail stent placed after LAMS removal	26 (93%)	6 (100%)	>0.99
Duration of LAMS (days), median (IQR)	35 (28, 49)	30 (27, 56)	0.90

**Table 3 diagnostics-13-01933-t003:** Outcome comparison of EUS-GBD for AC versus NC.

	Acute Cholecystitis (*n* = 39)	Non-Cholecystitis (*n* = 12)	*p* Value
Technical success	36 (92%)	11 (92%)	>0.99
Clinical success	34 (94%)	11 (100%)	>0.99
Procedure duration (minutes), median (IQR)	43 (25, 52)	45 (26.5, 89.5)	0.37
Post-procedure length of stay (days), median (IQR)	3 (2, 6)	3 (1.5, 7.5)	0.97
Total gallbladder procedures, median (IQR)	2 (2, 2)	2 (1, 2)	0.59
Adverse event rate	4 (10%)	3 (25%)	0.33
Survival (days), median (IQR)	243 (49, 347)	202 (48, 287)	0.36
Follow up duration (days), median (IQR)	463 (92, 572)	131 (55, 394)	0.46

**Table 4 diagnostics-13-01933-t004:** Adverse events after EUS-GBD.

	Acute Cholecystitis (*n* = 39)	Non-Cholecystitis (*n* = 12)	*p* Value
**Adverse Events, *n* (%)**	4 (10%)	3 (25%)	0.33
**Early (<4 weeks)**			
Abdominal pain requiring admission	1 (2.6%)	-	
Bleeding requiring transfusion	1 (2.6%)	-	
Bleeding requiring embolization	1 (2.6%)	-	
Symptomatic pneumoperitoneum requiring exploratory laparotomy	-	1 (8.3%)	
Symptomatic pneumoperitoneum managed with supportive care	-	1 (8.3%)	
Post-procedure respiratory failure requiring mechanical ventilation	-	1 (8.3%)	
**Late (>4 weeks)**			
Cholecystocolonic fistula	1 (2.6%)	-	

## Data Availability

The data are not publicly available due to privacy and ethical concerns.

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
