# Peer review of "Endoscopic Ultrasound-Guided Gallbladder Drainage: Beyond Cholecystitis"

_diagnostics, 2023, doi:10.3390/diagnostics13111933_

Round 1

Reviewer 1 Report

This is an interesting and sound retrospective study. The main limitation is the small sample size in the NC group, which the authors acknowledge in the discussions section. 

Author Response

Reviewer 1 left no comments to address.

Reviewer 2 Report

The authors evaluated the clinical outcomes of EUS-GBD for AC and NC indications.

They presents EUS-GBD for NC is as safe and effective as EUS-GBD for AC.

This is a well written, interesting paper, and useful contributing to clinical practice.

However, I would like to suggest some issues with several comments and criticisms as follows;

Comment 1:

Materials and Methods session:

The definition of “non-surgical patients” is unclear.

The authors needs to clarify the criteria of “non-surgical patients”.

Comment 2:

Materials and Methods session:

Because the background diseases are different between AC group and NC group, the contents of clinical success of EUS-GBD are different in two groups. Therefore, it is not appropriate to define clinical success as primary endpoint, in this study. Study outcomes and definitions need to be reconsidered

Comment 3:

Results session:

In this study, LAMS were removed in 78% of AC patients compared to 55% in NC patients. The authors needs to describe the criteria and procedure methods of LAMS removal.

Comment 4:

Results session:

In this study, DPT stent was replaced in 93% of AC patients and 100% of NC patients.

Is this attempted to be placed in all cases when LAMS is removed?

The authors needs to clarify this point and to add the details of unsuccessful case of DPT placement in AC patients.

Comment 5:

Results session:

There are several cases of continuous LAMS placement in both groups. However, this study did not evaluate late adverse events of continued LAMS placement, other than death events. The authors needs to describe the late adverse events of continued LAMS placement.

Comment 6:

Results session: Page 7, line 9

“The standard practice of stone extraction prior to LAMS removal” seems to be a very interesting procedure. The authors needs to describe the procedure method in detail.

Comment 7:

Results session: Adverse Events

If possible, I think it would be better if you could also add a table for AEs comparison between the two groups.

Comment 8:

Abstract and Results session:

The “p-value” of AEs in the two groups are different; Abstract is “p=0.62”, Results session is “p=0.33”. Please revise it correctly.

Round 2

Reviewer 2 Report

The author has responded appropriately to all comments.

For Table4, please add the frequency of adverse events and P-value for both groups.

Author Response

Thank you for your comment.  Table 4 has been adjusted accordingly and the revised manuscript has been uploaded.